# Deep Reinforcement Learning in Agent Based Financial Market Simulation

**Iwao Maeda** [1,*]**, David deGraw** [2]**, Michiharu Kitano** [3]**, Hiroyasu Matsushima** [1] 🄳**, Hiroki Sakaji** [1]**, Kiyoshi Izumi** [1] **and Atsuo Kato** [3]

[1]  Department of Systems Innovation, School of Engineering, The University of Tokyo, Tokyo 113-8654, Japan; matsushima@sys.t.u-tokyo.ac.jp (H.M.); sakaji@sys.t.u-tokyo.ac.jp (H.S.); izumi@sys.t.u-tokyo.ac.jp (K.I.)
[2]  Daiwa Securities Co. Ltd., Tokyo 100-0005, Japan; david.degraw@daiwa.co.jp
[3]  Daiwa Institute of Research Ltd., Tokyo 135-8460, Japan; michiharu.kitano@dir.co.jp (M.K.); atsuo.kato@dir.co.jp (A.K.)
*  Correspondence: d2018imaeda@socsim.org

**Abstract:** Prediction of financial market data with deep learning models has achieved some level of recent success. However, historical financial data suffer from an unknowable state space, limited observations, and the inability to model the impact of your own actions on the market can often be prohibitive when trying to find investment strategies using deep reinforcement learning. One way to overcome these limitations is to augment real market data with agent based artificial market simulation. Artificial market simulations designed to reproduce realistic market features may be used to create unobserved market states, to model the impact of your own investment actions on the market itself, and train models with as much data as necessary. In this study we propose a framework for training deep reinforcement learning models in agent based artificial price-order-book simulations that yield non-trivial policies under diverse conditions with market impact. Our simulations confirm that the proposed deep reinforcement learning model with unique task-specific reward function was able to learn a robust investment strategy with an attractive risk-return profile.

**Keywords:** deep reinforcement learning; financial market simulation; agent based simulation

## 1. Introduction

In recent years, applications of deep learning to predicting financial market data have achieved some level of success (Chong et al. 2017; Long et al. 2019; Lahmiri and Bekiros 2019, 2020). However, issues such as heteroskedasticity Nelson (1991), low signal-to-noise Aaker and Jacobson (1994), and the large observer effect seen in market impact, make their use in real-world applications challenging. Furthermore, the predictive ability of deep learning models is highly dependent on the data used in training, and for samples with "unknown" features (also known as out-of-distribution samples (Lee et al. 2018)), these models can make surprisingly nonsensical predictions and is a critical issue when applying deep learning models to real world tasks (Sensoy et al. 2018).

In financial areas, there are models for providing automated financial advice or investment management called robo-advisors (Leong and Sung 2018; Sironi 2016). Robo-advisors predict future market states and determine actions according to their predictions. Therefore, poor predictions could potentially lead to unacceptable levels of risk and large losses of capital. As a common financial disclaimer states that "past performance is not a guarantee

of future returns", the fundamental difficulty is that past data is often a poor predictor of the future (Silver 2012), and complex dynamical properties of financial markets (Arthur 1999) make it difficult to incorporate prior knowledge about the target distribution. Financial practitioners have traditionally been limited to training models with past data and do not have many options for improving their predictive models (Bailey et al. 2014). Moreover, such "back-testing" of models cannot account for transaction costs or market impact, both of which can be of comparable magnitude to the forecast returns (Hill and Faff 2010).

In order to overcome such issues, we argue that artificial market simulation provides a promising avenue for improving the predictive ability of deep reinforcement learning models. Agent-based artificial market simulation is an established and widely studied methodology (Raberto et al. 2001; Streltchenko et al. 2005) that has been shown to be of practical significance as alternatives to real markets (Brewer et al. 2013; Muranaga et al. 1999; Raberto et al. 2001). One of the main advantages of simulated markets is that they can be adapted to create realistic scenarios and regimes that have never been realized in the past (Lux and Marchesi 1999; Silva et al. 2016). Multi-agent simulation is a system of environment and agents where the actions of agents are informed by the environment and the state of the environment evolves, in turn, through the actions of the agents. Typically, in financial simulations, the market is designated as the environment where assets are transacted by an ensemble of agents modeled after investors and whose behavior is governed by reductive formulas.

Efforts to learn deep reinforcement learning (DRL) strategies on environmental simulators have been successful in various domains (Gu et al. 2017; Gupta et al. 2017; Li et al. 2016) and shown to attain or exceed human-level performance on highly complex reward-maximizing gameplay tasks (Mnih et al. 2013, 2015) such as GO and StarCraft II (Vinyals et al. 2019). Deep reinforcement learning agents base their actions on predictions about the environment, and train their networks to maximize their cumulative rewards obtained through their individual actions. As mentioned previously, models trained in simulation with DRL are understood to derive their superior predictive ability from having trained across more different scenarios than can be experienced by any one human in a lifetime.

Another advantage of agent simulations is that DRL models can be trained in an environment with realistic transaction costs and market impact (Donier et al. 2015). Market impact is the influence of your own investment actions on market states and is understood to adversely affect investment returns. Market impact is something that cannot be reproduced from historical data alone since the relationships between actions and effects cannot be replicated in such back-testing.

Previous simulation and deep reinforcement learning research in finance (Raman and Leidner 2019; Ritter 2018; Spooner et al. 2018) has been limited by overly simplistic agents with a limited action space. In contrast, our proposed agent based framework for training DRL agents yield sophisticated trading strategies that achieve robust risk-return profiles under diverse conditions with market impact and transaction costs.

The main contributions of our current work are as follows:

1. Proposal and verification of a deep reinforcement learning framework that learns meaningful trading strategies in agent based artificial market simulations
2. Effective engineering of deep reinforcement learning networks, market features, action space, and reward function.

## 2. Related Work

### 2.1. Stock Trading Strategies

Stock trading strategies have been studied for a long time (LeBaron 2002). Traditionally, dynamic model-based approaches apply simple rules and formulas to describe trader behavior, such as the sniping strategy (Rust et al. 1992), the Zero intelligence strategy (Ladley 2012), and the risk-based bidding strategy

(Vytelingum et al. 2004). In recent years, reinforcement learning (RL) methods (Dayan and Balleine 2002; Sutton et al. 1998), especially deep reinforcement learning (DRL) methods (Mnih et al. 2013, 2015) have been applied to learning investment strategies (Meng and Khushi 2019; Nevmyvaka et al. 2006), such as DRL for financial portfolio management (Jiang et al. 2017), market making via reinforcement learning (Spooner et al. 2018), and DRL for price trailing (Zarkias et al. 2019).

### 2.2. Deep Reinforcement Learning

DRL (as well as RL) is roughly classified into two types—value based and policy based approaches. Value based deep reinforcement learning methods (Littman 2001) approximate value functions called Q-functions using deep neural networks, and actions with maximum Q-functions are selected. Rainbow (Hessel et al. 2018) claims that combination of major methodologies in DRL, such as double DQN (Van Hasselt et al. 2016) and dueling network (Wang et al. 2015) improves performance of DQN drastically. General reinforcement learning architecture (Gorila) (Nair et al. 2015) provided a parallel training procedure for fast training. Ape-X (Horgan et al. 2018) proposed an algorithm for distributed processing of prioritized experience replay (Schaul et al. 2015). On the other hand, policy based deep reinforcement learning methods (Sutton et al. 2000) approximate the optimal policy directly using deep neural networks. Especially, actor-critic methods which train actor and critic functions simultaneously have been intensely studied, such as asynchronous advantage actor-critic (A3C) (Mnih et al. 2016), deep deterministic policy gradient (DDPG) (Silver et al. 2014), and trust region policy optimization (TRPO) (Schulman et al. 2015).

DRL is applied to various tasks such as table games (Silver et al. 2016, 2017) video games (Lample and Chaplot 2017; Mnih et al. 2013), autonomous driving (Pan et al. 2017; Sallab et al. 2017), and robotic manipulation (Gu et al. 2017; Kalashnikov et al. 2018).

### 2.3. Financial Market Simulation

Financial market simulation has been used for investigating market microstructure (Muranaga et al. 1999) and financial market regulations (Mizuta 2016). In particular, multi agent financial market simulation (LeBaron et al. 2001; Lux and Marchesi 1999; Samanidou et al. 2007) is commonly used. In early agent based simulations, such as the rebalancers and portfolio insurers model (Kim and Markowitz 1989) and the econophysics approach (Levy et al. 1994), were not able to have realistic time series characteristics called stylized facts (Harvey and Jaeger 1993; Levine and Demirgüç-Kunt 1999). In subsequent studies, however, stylized facts have been observed by approaches such as application of percolation theory (Solomon et al. 2000; Stauffer 2001), and fundamentalist and chartist model (Lux and Marchesi 1999). In this study, virtual markets are created using such a simulation theory.

## 3. Simulation Framework Overview

The overview of our proposed framework is shown in Figure 1. The simulator consists of markets and agents where markets play the role of the environment whose state evolves through the actions of agents. Each agent, in turn, decides its action according to observations of the markets' states. We incorporate a degree of randomness in the agents' selection of actions and the objective of the agents is to maximize their capital amount *cap* calculated with the following equation.

$$cap = cash + \sum_i p_{\text{Mid},i} pos_i,\tag{1}$$

where *cash*, $p_{\text{Mid},i}$, and $pos_i$ are the amount of cash, the midprice of instrument *i*, and quantity of instrument *i*, respectively. In each simulation step, we sample from a distribution of agents until we find

one that submits an order. Subsequently, the market order book is updated with the submitted order. The parameters of the DRL agent model is trained at a fixed time intervals of the simulation.

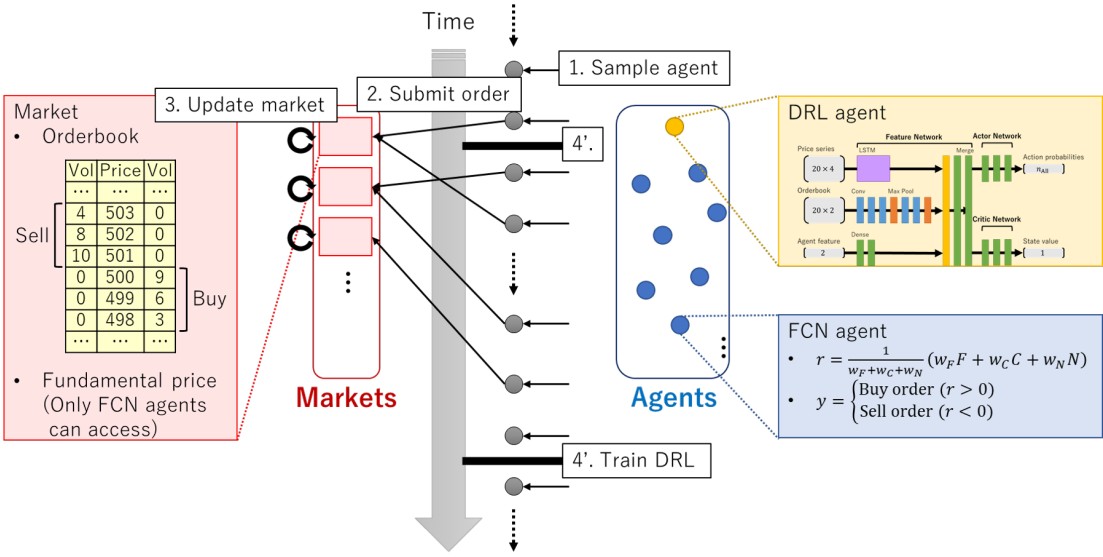

**Figure 1.** Overview of the simulation framework proposed in this research. The simulator consists of markets and agents where markets play the role of the environment whose state evolves through the actions of the agents. There are two types of agents—deep reinforcement learning (DRL) agent and fundamental-chart-noise (FCN) agents. The objective of the agents is to maximize their capital amount. In each step of simulation, an agent to submit an order is sampled, the agent submits an order, and markets process orders and update their orderbooks. The DRL model in the DRL agent is trained in a certain time intervals.

## 4. Simulator Description

### 4.1. Markets

Our simulated orderbook market consists of an instrument, prices, quantities, and side (buy or sell). An order must specify these four properties as well as an order type from the below three:

- Limit order (LMT)
- Market order (MKT)
- Cancel order (CXL)

The exceptions are that MKT orders do not need to specify a price, and CXL orders do not need to specify quantity since we do not allow volume to be amended down in our simulation (i.e., CXL orders can only remove existing LMT or MKT orders).

Market pricing follows a continuous double auction Friedman and Rust (1993). A transaction occurs if there is a prevailing order on the other side of the orderbook at a price equal to or better than that of the submitted order. If not, the order is added to the market orderbook. When there are multiple orders which meet the condition, the execution priority is given in order of price first, followed by time. CXL orders remove the corresponding limit order from the orderbook, and fails if the target order has already been executed.

The transaction volume $v$ is determined by:

$$v = \min(v_{\text{buy}}, v_{\text{sell}}), \tag{2}$$

where $v_{buy}$ and $v_{sell}$ are submitted volumes of buy and sell orders.

After $v$ is determined, stock and cash are exchanged according to the price and transaction volume between the buyer and seller. Executed buy and sell orders are removed from the orderbook if $v = v_{buy}$ or $v = v_{sell}$, and change to volume $v_{buy} - v$ or $v_{sell} - v$ otherwise.

Additionally, we define a fundamental price $p_F$ for each market. The fundamental price represents the fair price of the asset/market and observable only by FCN agents (not the DRL agents) and is used to predict future prices. The fundamental price changes according to a geometric Brownian motion (GBM) Eberlein et al. (1995) process.

Markets required the following hyperparmeters:

- Tick size
- Initial fundamental price
- Fundamental volatility

Tick size is the minimum price increment of the orderbook. The initial fundamental price and fundamental volatility are parameters of the GBM used to determine the fundamental price.

### 4.2. Agents

Agents registered to the simulator are classified into the following two types:

- Deep Reinforcement Learning (DRL) agent
- Fundamental-Chart-Noise (FCN) agent

The DRL agent is the agent we seek to train, while the FCN agents comprise the environment of agents in the artificial market. Details of the DRL agent are described in Section 5.1.

The FCN agent (Chiarella et al. 2002) is a commonly used financial agent and predicts the log return $r$ of an asset with a weighted average of fundamental, chart, and noise terms.

$$r = \frac{1}{w_F + w_C + w_N}(w_F F + w_C C + w_N N).$$ (3)

Each terms is calculated by the following equations below. The fundamental term $F$ represents the difference between the reasonable price considered by the agent and the market price at the time, the chart term $C$ represents the recent price change, and the noise term $N$ is sampled from a normal distribution.

$$F = \frac{1}{T}\log(\frac{p_t^*}{p_t})$$ (4)

$$C = \frac{1}{T}\log(\frac{p_t}{p_{t-\tau}})$$ (5)

$$N \sim \mathcal{N}(\mu, \sigma^2).$$ (6)

$p_t$ and $p_t^*$ are current market price, fundamental price, and $\tau$ is the time window size, respectively. $p_t^*$ changes according to a geometric Brownian motion (GBM). Weight values $w_F$, $w_C$, and $w_N$ are independently random sampled from exponential distributions for each agent. The scale parameters $\sigma_F$, $\sigma_C$, and $\sigma_N$ are required for each simulation. Parameters of the normal distribution $\mu$ and $\sigma$ are fixed to 0 and 0.0001.

The FCN agents predict future market prices $p_{t+\tau}$ from the predicted log return with the following equation:

$$p_{t+\tau} = p_t \exp(r\tau).$$ (7)

The agent submits a buy limit order with price $p_{t+\tau}(1-k)$ if $p_{t+\tau} > p_t$, and submits a sell limit order with price $p_{t+\tau}(1+k)$ if $p_{t+\tau} < p_t$. The parameter $k$ is called order margin and represents the amount of profit that the agent expects from the transaction. The submitting volume $v$ is sampled from a discrete uniform distribution $u\{1,5\}$. In order to control the number of outstanding orders in the market orderbook, each order submitted by the FCN agents has a time window size, after which the order is automatically canceled .

### 4.3. Simulation Progress

Simulation proceeds by repeating the order submission by the agents, order processing by the markets, and updating the fundamental prices at the end of each step. The first 1000 steps are pre-market-open steps used to build the market orderbook and order processing is not performed.

Each order action consists of order actions by the FCN and DRL agents. FCN agents are randomly selected and given a chance to submit an order. The FCN agent submits an order according to the strategy described in Section 4.2 with probability 0.5 and does nothing with probability 0.5. If the FCN agent submits an order, the order is added to the orderbook of the corresponding market. Similarly, an actionable DRL agent is selected, and the DRL agent submits an order according to the prediction made from observing the market state. The DRL agent may act again after an interval sampled from a normal distribution $\mathcal{N}(100, 10^2)$.

Once an agent submits an order, the market processes the order according to the procedure described in Section 4.1. After processing the orders, each market deletes orders that have been posted longer than the time window size of the relevant agent.

In the final step, each market updates its fundamental price according to geometric Brownian motion.

Additionally, training of the DRL agent is performed at a fixed interval. The DRL agent collects recent predictions and rewards from the simulation path and updates its policy gradients.

## 5. Model Description

### 5.1. Deep Reinforcement Learning Model

DRL uses deep learning neural networks with reinforcement learning algorithms to learn optimal policies for maximizing an objective reward. In this study, deep reinforcement learning models are trained to maximize the financial reward, or returns, of an investment policy derived from observations of, and agent actions in, a simulated market environment.

Various types of deep reinforcement learning methods are used in financial applications depending on the task (Deng et al. 2016; Jiang and Liang 2017; Jiang et al. 2017). In this study, we used advantage-actor-critic (A2C) network (Mnih et al. 2016). An A2C network is a version of actor-critic network (Konda and Tsitsiklis 2000), and has two prediction paths, one for the actor and one for the critic network. The actor network approximates and optimal policy $\pi(a_t|s_t;\theta)$ (Sutton et al. 2000), while the critic network approximates the state value $V(s_t;\theta_v)$, respectively. The gradient with respect to the actor parameters $\theta$ takes the form $\nabla_\theta \log(\pi(a_t|s_t;\theta))(R_t - V(s_t;\theta_v)) + \beta\nabla_\theta H\pi(a_t|s_t;\theta))$, where $R_t$ is the reward function, $H$ is the entropy and $\beta$ is the coefficient. The gradient with respect to the critic parameters $\theta_v$ takes the form $\nabla_{\theta_v}(R_t - V(s_t;\theta_v))^2$.

### 5.2. Feature Engineering

The price series is comprised of 20 contiguous time steps taken at 50 time step intervals of four market prices—the last (or current) trade price, best ask (lowest price in sell orderbook), best bid (highest price in buy orderbook), and mid price (average of best ask and best bid). Each price series is normalized by the price values of the first row.

The orderbook features are arranged in a matrix of the latest orderbook and summarize order volumes of upper and lower prices centered at the mid price. Order volumes of all agents and the predicting agent are aggregated at each price level. To distinguish buy and sell orders, buy order volumes are recorded as negative values. The shape of each orderbook feature is $20 \times 2$.

The agent features consist of cash amount, and stock inventory of each market. The shape of the agent feature is $1 + n_{\text{Market}}$, where $n_{\text{Market}}$ is the number of markets.

*5.3. Actor Network*

The actor network outputs action probabilities. Each action has the following five parameters:

- Side: [Buy, Sell, Stay]
- Market: $[0, 1, \ldots]$
- Type: [LMT, MKT, CXL]
- Price: [0, 2, 4, 6, 8, 10]
- Volume: [1, 5]

Side indicates the intent of the order. When side is "Stay", the other four parameters are ignored and no action is taken. Market indicates the target market. Price is the difference of submitting price from the best price (i.e., depth in the orderbook), and the submitting price $p$ is calculated by the following equation:

$$p = \begin{cases} p_{\text{BestAsk}} - \text{Price} & (\text{Action} = \text{Buy}) \\ p_{\text{BestBid}} + \text{Price} & (\text{Action} = \text{Sell}). \end{cases} \tag{8}$$

Both price and volume are used when type is LMT, and only volume is used when type is MKT. When type is CXL, we allow whether to cancel the order with highest price or lowest price. Finally, the number of all actions $n_{\text{All}}$ is calculated as:

$$n_{\text{All}} = 2n_{\text{Market}}(n_{\text{Price}}n_{\text{Volume}} + n_{\text{Volume}} + 2) + 1, \tag{9}$$

where $n_{\text{Market}}$, $n_{\text{Price}}$, and $n_{\text{Volume}}$ indicate the number of markets, price categories, and volume categories, respectively.

Agents select actions by roulette selection with the output probability. Additionally, DRL agents perform random action selection with probability $\epsilon$ according to the epsilon greedy strategy (Sutton and Barto 2018).

*5.4. Reward Calculation*

A typical objective for financial trading agents is to maximize their capital amount calculated by the following equation.

$$cap = cash + \sum_i p_{\text{Mid},i} pos_i. \tag{10}$$

The reward should then be some function that is proportional to the change in capital amount. For this study, two reward functions are used: the capital-only reward $R_{\text{CO}}$, and the liquidation-value reward $R_{\text{LV}}$. $R_{\text{CO}}$ is calculated as the difference between the capital after investment $cap_a$ and the baseline capital $cap_b$. The baseline capital was calculated with amount of cash and inventory without investment.

$$R_{\text{CO}} = cap_{t+\tau} - cap_t \tag{11}$$

$$cap_{t+\tau} = cash_{t+\tau} + \sum_i p_{\text{Mid},i,t+\tau} pos_{i,t+\tau} \tag{12}$$

$$cap_t = cash_t + \sum_i p_{\text{Mid},i,t} pos_{i,t}, \tag{13}$$

where $t$ and $\tau$ are the time of action and the time constant, respectively.

On the other hand, $R_{\text{LV}}$ is the calculated as the difference between the liquidation value after investment $LV_a$ and the baseline liquidation value $LV_b$. The liquidation value is defined as the cash amount that would remain if the entire inventory was instantly liquidated. If the inventory cannot be liquidated within the constraints of the current market orderbook, the remaining inventory is liquidated with a penalty price (0 for long inventory and $2p_{\text{Market}}$ for short inventory).

$$R_{\text{LV}} = LV_{t+\tau} - LV_t. \tag{14}$$

The liquidation value $LV$ strongly correlates to *cap* but carries a penalty proportional to the magnitude of the inventory. The capital-only reward includes no penalty to the inventory value, and use of it may cause training of risky strategies (Fama and French 1993). We anticipate the use of $R_{\text{LV}}$ may be effective in training DRL strategies that balance inventory risk with trading reward.

In training, calculated rewards are normalized for each training batch with the following equation.

$$R_i' = \frac{R_i}{\sqrt{E[R^2]}}. \tag{15}$$

In addition, a penalty value $R_p$ is add to the normalized reward when the action selected by the DRL agent was infeasible at the action phase (inappropriate cancel orders etc.). $R_p$ was set to $-0.5$.

### 5.5. Network Architecture

Overview of the A2C network used in this research is shown in Figure 2. As the number of markets is 1 in this research, the length of agent features is 2. The network consist of three networks—a market feature network, an actor network, and a critic network. The market feature network uses a long short-term memory (LSTM) (Hochreiter and Schmidhuber 1997) layer to extract features from the market price series according to previous studies (Bao et al. 2017; Fischer and Krauss 2018), as well as convolutional neural network (CNN) layers to extract orderbook features that have positional information (Tashiro et al. 2019; Tsantekidis et al. 2017). The agent features are extracted by dense layers and the actor network outputs action probabilities while the critic network outputs predicted state values. ReLU activation is applied to the convolutional and dense layers in the network except for the last layers of actor and critic networks which use softmax activation so that the action probability outputs sum to unity.

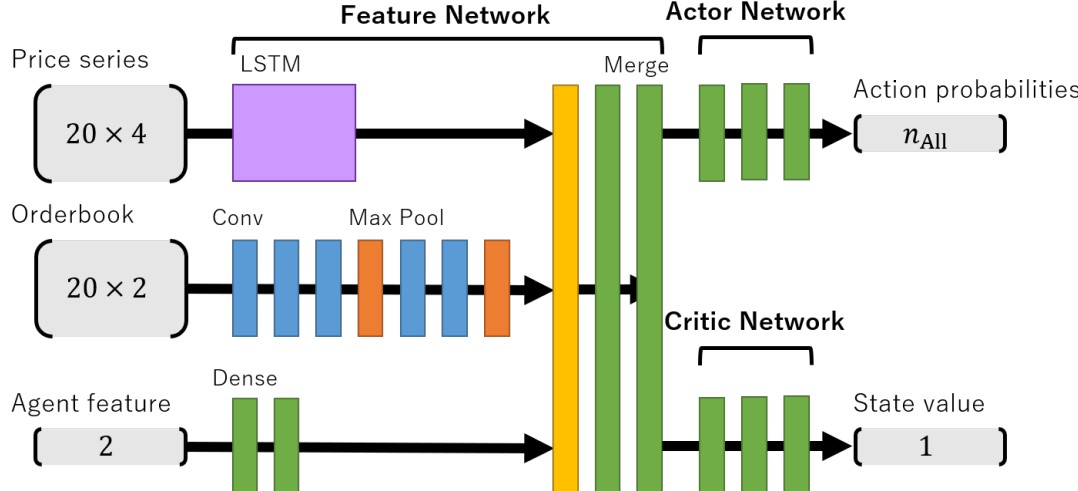

**Figure 2.** Overview of the A2C network used in this research ($n_{\text{market}} = 1$). The network takes price series, orderbook feature and agent feature as input variables and outputs action probabilities and state values. Purple, blue, orange, green, and yellow boxes in the network represent LSTM, convolutional, max pooling, dense, and merge layers, respectively. LSTM layers have tangent hyperbolic activation. Convolutional and dense layers except the last layers of actor and critic networks have ReLU activation. Action probabilities are calculated by applying softmax activation to the output of actor network.

## 6. Experiments

Experiments were performed using the simulator and deep reinforcement learning model described in previous sections. The purpose of the experiments is to investigate following questions:

1. Whether the model can learn trading strategies on the simulator
2. Whether the learned strategies are valid

DRL models with the capital-only (CO) and liquidation-value (LV) reward functions (the CO-DRL and LV-DRL models) were trained in one simulation and validated in a separate simulation. In the validation simulations, DRL models select actions only by their network outputs and do not perform random action selections. Each simulation consists of 100 sub simulations which have 1,000,000 steps (total 100,000,000 steps). Each sub simulation has different simulator and agent parameter settings. For comparison, a model that selects actions randomly (with the same probability) from the same action set as the DRL models was trained in the same simulations.

Model performances were compared along the following benchmarks:

- Average reward $\overline{R}$
- Sharpe ratio $S_p$
- Maximum drawdown *MDD*

Average reward is the average of all actions of the agent in simulation. Sharpe ratio $S_p$ (Sharpe 1994) is a metric for measuring investment efficiency and calculated by following equation:

$$S_p = \frac{E[R_a - R_b]}{\sqrt{\text{Var}[R_a - R_b]}},$$ (16)

where $R_a$ and $R_b$ are returns of the investment and benchmark, respectively. In this study, $R_b = 0$ was assumed. Maximum drawdown (MDD) *MDD* (Magdon-Ismail et al. 2004) is another metric used to measure investment performance and calculated by the following equation:

$$MDD = \frac{P - L}{P},$$

(17)

where $P$ and $L$ are the highest and lowest capital amount before and after the largest capital drop.

*6.1. Simulator Settings*

Some parameters of simulator and agents were fixed in all sub simulations, while others were randomly sampled in each sub simulation.

The values/distributions of market parameters are shown below.

- Tick size $p_{\min} = 1$
- Initial fundamental price $p_0^* \sim \mathcal{N}(500, 50^2)$
- Fundamental volatility $f \sim \mathcal{N}(0.0001, 0.00001^2)$

The values/distributions of parameters for FCN agents are shown below.

- Number of agents $n_{\text{FCN}} = 1000$
- Scale of exponential distribution for sampling fundamental weights $\sigma_C \sim \mathcal{N}(0.3, 0.03^2)$
- Scale of exponential distribution for sampling chart weights $\sigma_C \sim \mathcal{N}(0.3, 0.03^2)$
- Scale of exponential distribution for sampling noise weights $\sigma_N \sim \mathcal{N}(0.3, 0.03^2)$
- Time window size $\tau \sim u(500, 1000)$
- Order margin $k \sim u(0, 0.05)$

The values/distributions of parameters for DRL agent are shown below.

- Initial inventory of DRL agent $pos_0 \sim \mathcal{N}(0, 10^2)$
- Initial cash amount of DRL agent $cash_0 = 35000 - p_0 pos_0$

*6.2. Results*

Average rewards of each sub simulation in training and validation are shown in Figures 3 and 4. Both figures represents average rewards of random, CO-DRL, and LV-DRL models. As shown, average reward gradually improves as simulation progresses in both DRL models, but at a lower rate and overall magnitude in the CO-DRL model. Similarly in validation, the average rewards of the LV-DRL model are orders of magnitude higher than the other models across all sub simulations.

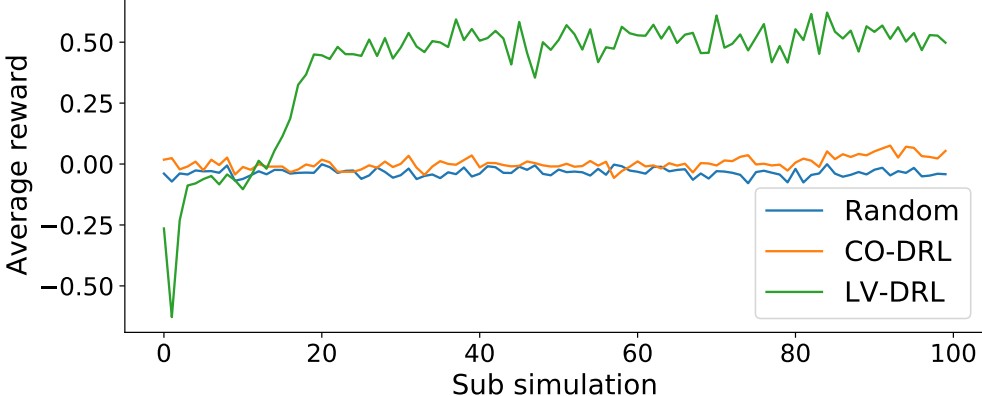

**Figure 3.** Average rewards of each sub simulation in training. Blue, orange and green lines represents average rewards of random, capital only-deep reinforcement learning (CO-DRL), and liquidation-value deep reinforcement learning (LV-DRL) models.

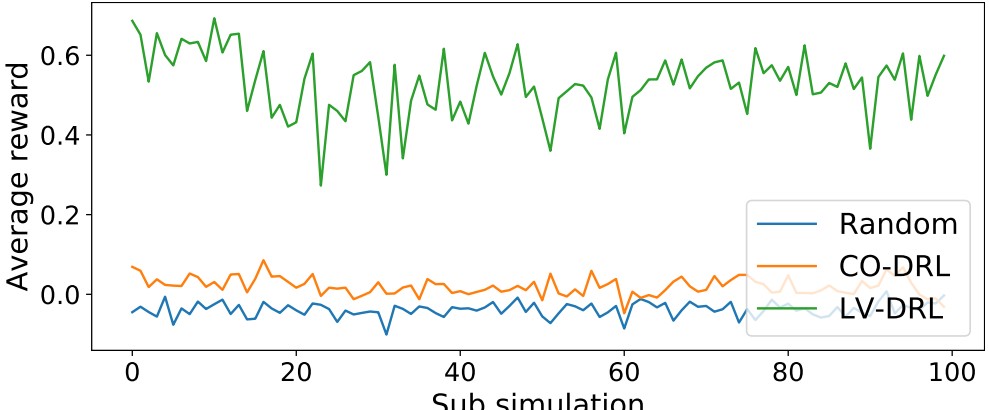

**Figure 4.** Average rewards of each sub simulation in validation. Orange and blue lines represents average rewards of deep reinforcement learning (DRL) and random (baseline) models.

Model performance in training and validation simulations are shown in Table 1. Table 1 shows the mean and standard deviation of the average rewards , Sharpe ratios, and maximum drawdowns across the sub simulations. The evaluation metrics clearly show that the LV-DRL agent outperforms the others across all benchmarks. We can see that the liquidation-value reward function was instrumental for the DRL agent to learn a profitable trading strategy that simultaneously mitigates inventory risk.

**Table 1.** Model performances of random, CO-DRL, and LV-DRL models. $\overline{R}$, $S_p$, and *MDD* are average reward, Sharpe ratio, and maximum drawdown. Indices in the table are mean and standard deviation of sub simulations.

| | **Training Simulation** | | | **Validation Simulation** | | |
|---|---|---|---|---|---|---|
| **Model** | $\overline{R} \uparrow$ | $S_p \uparrow$ | $MDD \downarrow$ | $\overline{R} \uparrow$ | $S_p \uparrow$ | $MDD \downarrow$ |
| Random | $-0.035 \pm 0.017$ | $0.005 \pm 0.005$ | $0.111 \pm 0.143$ | $-0.038 \pm 0.018$ | $0.005 \pm 0.007$ | $0.110 \pm 0.104$ |
| CO-DRL | $0.004 \pm 0.025$ | $0.009 \pm 0.006$ | $0.144 \pm 0.196$ | $0.020 \pm 0.023$ | $0.009 \pm 0.006$ | $0.124 \pm 0.178$ |
| LV-DRL | $0.402 \pm 0.236$ | $0.043 \pm 0.023$ | $0.049 \pm 0.137$ | $0.530 \pm 0.079$ | $0.053 \pm 0.019$ | $0.012 \pm 0.004$ |

The capital change during sub simulations in validation are shown in Figure 5 and illustrate important features of the LV-DRL model under different scenarios. In result (a), the capital of the LV-DRL model

continues to rise through sub simulation despite a falling market. We see the LV-DRL model tends to keep the absolute value of its inventory near 0, and that capital swings are smaller than other two models. Result (b) shows a case where the CO-DRL model slightly outperforms in accumulated capital, but we see that this strategy takes excessive inventory risk, causing large fluctuations in capital. In contrast, the LV-DRL model achieves a similar level of capital without large swings at a much lower level of inventory, leading to a higher Sharpe ratio.

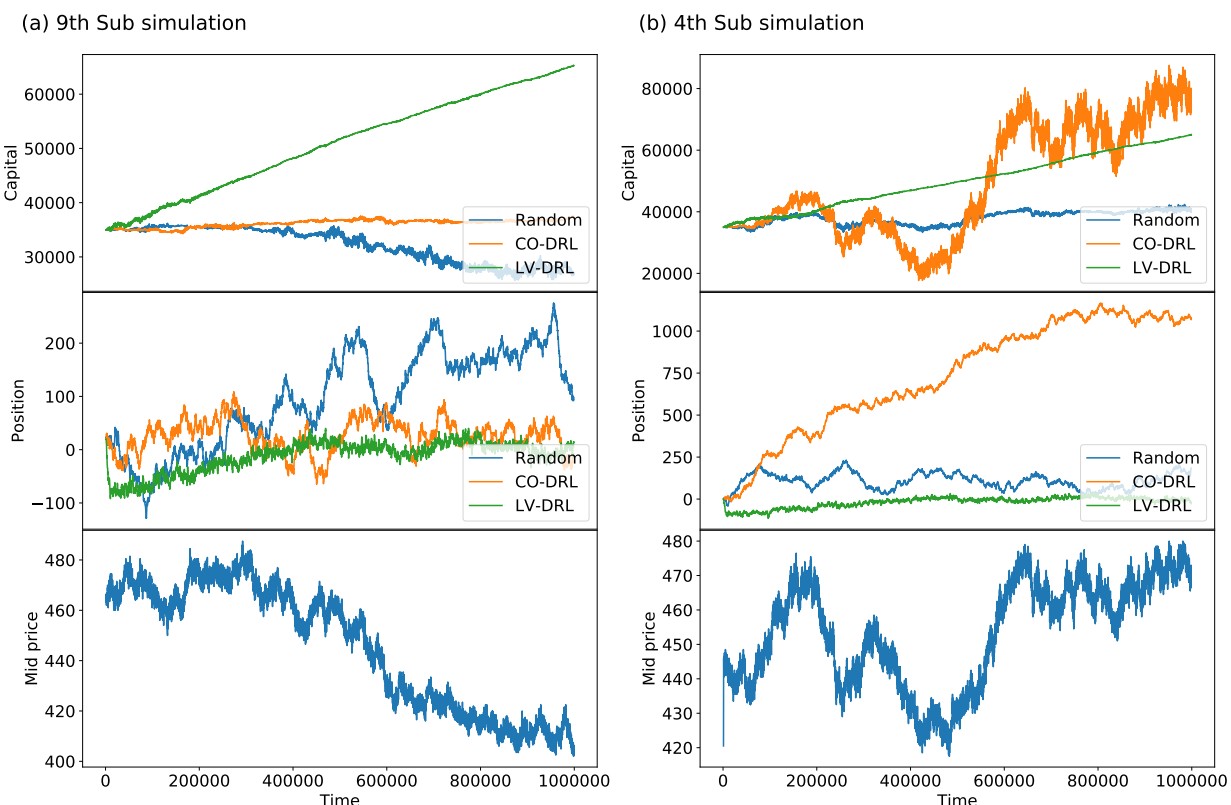

**Figure 5.** Example capital changes of sub simulations in validation. Both left and right columns show changes of capital, inventory, and mid price in one typical sub simulation.

A histogram of action probabilities of the three models in validation is shown in Figure 6. The horizontal axis shows 33 types of actions that can be selected by DRL agent. As explained in Section 5.3, actions of models are parameterized by five factors: Intent (Stay, Buy, Sell), Market, Order type (LMT, MKT, CXL), Price (difference from the base price in LMT and MKT orders, lowest or highest in CXL order), and Volume. We ignore the market factor since we only have one market in these experiments. The horizontal axis labels of Figure 6 actions are expressed by "Action-OrderType-Price-Volume".

As shown in Figure 6, the LV-DRL agent can be seen to avoid inefficient actions such as orders with unfavorable price and large volume, or cancel orders. The agent prefers to select buy and sell limit orders with price difference 4 and 6 and volume 5 while limit orders with price difference 0 and volume 5, market orders with volume 5, and cancel orders were rarely selected. Previous studies about high-frequency-trader and market making indicate that traders in real markets have similar investment strategies to our LV-DRL model (Hirano et al. 2019). On the other hand, since the reward function of the CO-DRL model does not consider the inventory of the agent, the CO-DRL model seems to pursue a momentum-trend-following strategy without much regard for risk. We also observe that the CO-DRL model submits more aggressive

market orders than the LV-DRL model; another indication that the CO-DRL agent disregards costs and risk in favor of capital gains.

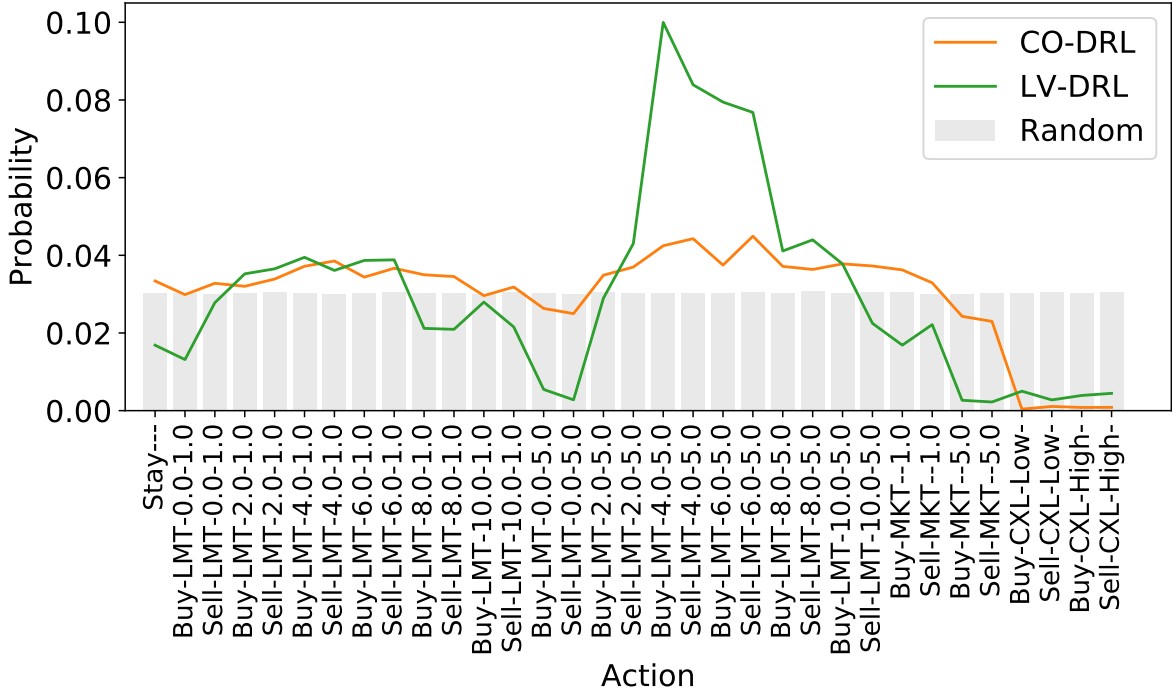

**Figure 6.** Action probabilities of three models in validation. The horizontal axis represents possible actions which are expressed by "Action-Order type-Price(difference from the price of market order in limit (LMT) and market (MKT) orders, lowest or highest in cancel (CXL) order)-Volume".

## 7. Conclusions

In this study we were able to show that with the appropriate reward function, deep reinforcement learning can be used to learn an effective trading strategy that maximizes capital accumulation without excessive risk in a complex agent based artificial market simulation. It was confirmed that the learning efficiency greatly differs depending on the reward functions, and the action probability distributions of well-trained strategies were consistent with investment strategies used in real markets. While it remains to be seen whether our proposed DRL model can perform in a real live financial market, our research shows that detailed simulation design can replicate certain features of real markets, and that DRL models can optimize strategies that adapt to those features. We believe that further consideration of realistic markets (such as multiple markets, agents with various behavioral principles, exogenous price fluctuation factors, etc.) will bring simulations closer to reality and enable creation of various DRL agents that perform well in the real world, and more advanced analyses of real markets.

One of the limitations of the current work is that there is only one DRL agent. In order to consider a more realistic market, we must simulate the interaction between various dynamic agents with differing reward functions. In future work, we plan to look at the effect of introducing multiple DRL agents and examining emergent cooperative and adversarial dynamics between the various DRL agents (Jennings 1995; Kraus 1997) and how that affects market properties as well as learned strategies.

**Author Contributions:** Conceptualization, I.M.; methodology, I.M., D.d., M.K., K.I., H.S. and A.K.; investigation, I.M.; resources, H.M. and H.S.; writing–original draft preparation, I.M. and D.d.; supervision, K.I.; project administration, K.I. and A.K. All authors have read and agreed to the published version of the manuscript.

**Funding:** This research received no external funding.

**Conflicts of Interest:** The authors declare no conflict of interest.

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
