# Peer review of "Deep Reinforcement Learning in Agent Based Financial Market Simulation"

_jrfm, doi:10.3390/jrfm13040071_

Round 1

Reviewer 1 Report

This is a very technical paper. Some suggestions on the application of the study should be provided at the conclusion. Here are some suggestions for consideration.

I consider the findings that has great value on Robo-dvisor. Therefore, the authors may consider to mention:

'Robo-advisor serves as financial adviser that provides automated financial advice or investment management for clients. Also, robo-advisor can provide personalized suggestions to clients in more effective ways, while the suggestions can also be updated according to real-time data'

Following two references could be used for the purpose.

Sironi, (2016) FinTech Innovation, 1st ed. Chichester, West Sussex, United Kingdom: Wiley, 2016

Leong, K. and Sung, A., (2018). FinTech (Financial Technology): what is it and how to use technologies to create business value in fintech way?. International Journal of Innovation, Management and Technology9(2), pp.74-78.

Reviewer 2 Report

  1. Remove references [1] and [2] which are not relevant and cite the following studies in Introduction:
  • Intelligent forecasting with machine learning trading systems in chaotic intraday Bitcoin market. Chaos, Solitons & Fractals 133, 109641, 2020.
  • Cryptocurrency forecasting with deep learning chaotic neural networks, Chaos, Solitons & Fractals 118, 35-40, 2020.
  • Deep learning networks for stock market analysis and prediction: Methodology, data representations, and case studies. Expert Systems with Applications Volume 8315 October 2017 Pages 187-205.
  • Deep learning-based feature engineering for stock price movement prediction. Knowledge-Based Systems Volume 16415 January 2019 Pages 163-173.

  1. Why using log-return knowing that artificial neural networks are capable to model both linear and nonlinear data? Argue.
  2. Are there are form of reward functions? Justify why using yours?
  3. Why using two different deep learning models to extract features (LSTM and CNN)? Why not using only CNN which more suitable to extract features not LSTM?
  4. What lessons can we learn from these simulations?
  5. What are the effects on prediction and profit generation on real data? Discuss.
